# Mechanisms Responsible for the Anticoagulant Properties of Neurotoxic *Dendroaspis* Venoms: A Viscoelastic Analysis

**DOI:** 10.3390/ijms21062082

**Published:** 2020-03-18

**Authors:** Vance G. Nielsen, Michael T. Wagner, Nathaniel Frank

**Affiliations:** 1Department of Anesthesiology, University of Arizona College of Medicine, Tucson, AZ 85719, USA; miketw01@anesth.arizona.edu; 2MToxins Venom lab LLC, 717 Oregon Street, Oshkosh, WI 54902, USA; nate@mtoxins.com

**Keywords:** anticoagulant activity, metalloproteinase, Kunitz-type inhibitor, three-finger toxin, thrombelastography, carbon-monoxide-releasing molecule

## Abstract

Using thrombelastography to gain mechanistic insights, recent investigations have identified enzymes and compounds in *Naja* and *Crotalus* species’ neurotoxic venoms that are anticoagulant in nature. The neurotoxic venoms of the four extant species of *Dendroaspis* (the Black and green mambas) were noted to be anticoagulant in nature in human blood, but the mechanisms underlying these observations have never been explored. The venom proteomes of these venoms are unique, primarily composed of three finger toxins (3-FTx), Kunitz-type serine protease inhibitors (Kunitz-type SPI) and <7% metalloproteinases. The anticoagulant potency of the four mamba venoms available were determined in human plasma via thrombelastography; vulnerability to inhibition of anticoagulant activity to ethylenediaminetetraacetic acid (EDTA) was assessed, and inhibition of anticoagulant activity after exposure to a ruthenium (Ru)-based carbon monoxide releasing molecule (CORM-2) was quantified. Black mamba venom was the least potent by more than two orders of magnitude compared to the green mamba venoms tested; further, Black Mamba venom anticoagulant activity was not inhibited by either EDTA or CORM-2. In contrast, the anticoagulant activities of the green mamba venoms were all inhibited by EDTA to a greater or lesser extent, and all had anticoagulation inhibited with CORM-2. Critically, CORM-2-mediated inhibition was independent of carbon monoxide release, but was dependent on a putative Ru-based species formed from CORM-2. In conclusion, there was great species-specific variation in potency and mechanism(s) responsible for the anticoagulant activity of *Dendroaspis* venom, with perhaps all three protein classes—3-FTx, Kunitz-type SPI and metalloproteinases—playing a role in the venoms characterized.

## 1. Introduction

While the coagulopathic effects of hemotoxic venoms derived from venomous snakes have been of great scientific interest for several decades [1], it is only recently that investigation has intensified on the effects of neurotoxic venoms on coagulation [2,3,4,5,6,7]. Specifically, utilizing the thrombelastograph, the effects of neurotoxic venoms containing proteolytic and lipolytic enzymes on human plasma-based coagulation or isolated human thrombin–fibrinogen systems have been defined in venoms obtained from snakes within the Elapidae and Viperidae families [2,3,4,5,6,7]. Of interest, the venoms containing neurotoxic phospholipase A_2_ (PLA_2_) that were investigated were found to have their anticoagulant effects inhibited by either specific phospholipase A_2_ inhibitors or tricarbonyldichlororuthenium (II) dimer (CORM-2) [3,4,5,6,7]. Critically, these preliminary investigations exploited the ability to assess changes in the hemostatic effects of enzyme-based neurotoxins in response to inhibitor exposure. Thus, the use of thrombelastography to detect neurotoxin activity via biochemical substrate nexuses of plasmatic coagulation with neurochemistry was conceived.

Although the anticoagulant properties of the neurotoxic venoms of several of *Naja* [2,3,4,5,6] and one *Crotalus* [7] species have been studied, another genus, *Dendroaspis* (the mambas), that kill their prey and humans with neurotoxic venom [8,9,10], were found to possess venom that was anticoagulant in vitro over 50 years ago [11,12]. Using the clotting-based, antiquated technology that was available in the 1960s, these investigators proposed that thrombin generation was impaired, fibrinogen was digested, fibrinolysis was impaired, and platelet aggregation decreased in blood exposed to Black Mamba (*Dendroaspis polylepis*), Eastern Green Mamba (*Dendroaspis angusticeps*) or Jameson’s Green Mamba (*Dendroaspis jamesoni*) venom [11,12]. The fourth extant species, Hallowell’s Green Mamba (*Dendroaspis viridis*), was not investigated [11,12]. Of interest, the venom of *D. polylepis* appeared to be between one and two orders of magnitude less potent as an anticoagulant compared to the other two species tested [11,12]. Critically, the mechanisms responsible for the observed anticoagulant activity in terms of venom compound or enzymes were not addressed by these studies or any subsequent works [11,12]. Recent characterizations of the proteome of venom obtained from the four species of *Dendroaspis* [13,14,15] offered differences that could potentially explain these anticoagulant potency differences [11,12]. As displayed in Table 1, unlike most venoms derived from Elapidae snakes, *Dendroaspis* venoms are predominantly composed of non-enzymatic neurotoxins such as 3-finger toxins (3-FTx) and Kunitz-type serine protease inhibitors (Kunitz-type SPI), with metalloproteinases (SVMPs) comprising less than 7% [13,14,15] and PLA_2_ comprising from zero to less than 0.15% [13] of the proteomes. As *D. polylepis* venom has a greater amount of Kunitz-type SPI than 3-FTx compared to the other three species, and given that isolated 3-FTx have acted as plasmatic anticoagulants [16,17], perhaps the predominance of 3-FTx in the other mamba species’ venom that was tested could explain the differences in anticoagulant potency [11,12]. Nevertheless, given that the vast majority of enzymes in the coagulation cascade are serine proteases, the possibility that key Kunitz-type SPI may play a role in venom-mediated anticoagulation should not be discounted. Lastly, given that mamba venom demonstrated fibrinogenolytic activity in these older investigations [11,12], the role played by metalloproteinases, even at the small percentages observed (Table 1), in anticoagulant activity must be considered. Considered in composite, a few key proteins from these two non-enzymatic protein classes in combination with metalloproteinases could account for the venom anticoagulant activity and differences in anticoagulant potency observed in *Dendroaspis* species.

Considering the above, the present investigation had the following goals. First, the anticoagulant effects of the venom obtained from these four species was to be characterized in human plasma via thrombelastography. Second, the contribution of metalloproteinases to the anticoagulant effects of the venoms was to be discerned by exposing them in isolation to the inhibitor ethylenediaminetetraacetic acid (EDTA) prior to thrombelastographic analysis. Third, the determination of inhibition of the anticoagulant activities of these venoms by exposure of venom in isolation to CORM-2 was to be performed. Fourth, the contribution of a putative ruthenium (Ru) radical formed during the release of carbon monoxide from CORM-2 towards the inhibition of venom anticoagulant activity was performed as previously described [18]. Such determinations of the mechanism responsible for the CORM-2-mediated inhibition of snake venom hemotoxicity is of interest considering its in vitro efficacy against numerous species’ venoms or isolated venom enzymes, as previously noted [4,5,6,7,19].

## 2. Results

### 2.1. Assessment of the Anticoagulant Activity of Dendroaspis Polylepis Venom in Human Plasma and Determination of the Inhibitory Effects of EDTA and CORM-2 on Venom Anticoagulant Activity Assessed with Thrombelastography

The effects of *D. polylepis* venom on human whole-blood and plasmatic coagulation were originally described with concentrations as great as 2.5 mg/mL [12], but given that the standard venom stock solution we use has a maximum concentration of 50 mg/mL, the maximum concentration possible would be 500 µg/mL in our system [4,5,6,7,19]. This is the case as the venom solution used is a 1% addition to the plasma mix used in our thrombelastographic system [4,5,6,7,19]. The thrombelastographic paradigm used was performance-based, with the criteria that qualifies a venom concentration to be used in experimentation, such that there were no samples with coagulation parameter values within a normal range, and that the suppression of coagulation be at least 50% of normal parameter values [4,5,6,7,19]. The initial concentrations used (*n* = 2–4 per concentration) were 0, 100, 200 and 500 µg/mL. This preliminary evaluation demonstrated that the concentration of 500 µg/mL of *D. polylepis* venom increased time to maximum thrombus generation (TMRTG, minutes—time to commencement of the maximum velocity of coagulation), decreased maximum rate of thrombus generation (MRTG, dynes/cm^2^/sec—the maximum velocity of clot growth) and decreased total thrombus generation (TTG, dynes/cm^2^—an assessment of clot strength) without any of the samples having normal coagulation. Thus, the concentration of venom used for all subsequent experimentation was 500 µg/mL, as displayed in Figure 1.

The venom was subsequently exposed to 5 mM EDTA (1% addition to venom solution) for 30 min at 37 °C prior to placement into plasma for thrombelastographic analysis in order to assess the extent that metalloproteinases contributed to anticoagulant activity. As can be noted in Figure 1, EDTA exposure had no significant effect on the anticoagulant activity of this venom. Lastly, it was decided to expose this venom to the maximum concentration of CORM-2 used by us (1 mM) for 5 min prior to placement in plasma for thrombelastographic analysis. Again, CORM-2 did not significantly diminish the anticoagulant activity of the venom. Instead, CORM-2 exposure seemed to enhance the venom-mediated decrease in MRTG values. These results are depicted in Figure 1. In sum, *D. polylepis* venom displayed the weakest anticoagulant activity of any venom assessed by this laboratory to date, but the concentration used by us was well within the range reported previously [12]. Lastly, a role for metalloproteinase activity as a component of this venom’s anticoagulant was discounted, leaving the possibility that either the 3-FTx and/or Kunitz-type SPI were responsible for the anticoagulant activity observed.

### 2.2. Assessment of the Anticoagulant Activity of Dendroaspis Angusticeps Venom in Human Plasma and of the Inhibitory Effects of EDTA and CORM-2 on Venom Anticoagulant Activity Assessed with Thrombelastography

Previous work had demonstrated the anticoagulant effects of *D. angusticeps* venom at concentrations as small as 1–4 µg/mL and fibrinogenolytic activity at a concentration of 1 mg/mL [11,12]. In our plasma-based system, it was found during our concentration-response trials that a concentration of 3 µg/mL was required to compromise coagulation but not abolish it. Specifically, at a concentration of 2 µg/mL, there was no detectable compromise of coagulation; in contrast, at 4 µg/mL, there was no detectable coagulation. As can be seen in Figure 2, this venom at the indicated concentration did not significantly affect TMRTG values, but it did significantly decrease MRTG and TTG values compared to control conditions. After exposure to EDTA in isolation, the anticoagulant effects of the venom were significantly and completely abrogated, indicating that a metalloproteinase was responsible for the observed anticoagulant properties. Exposure of this venom to our lowest concentration of CORM-2 (100 µM) in isolation in phosphate-buffered saline (PBS) resulted in a loss of its anticoagulant activity, with TMRTG, MRTG and TTG values of this condition not significantly different from control conditions. Lastly, exposure of the venom to CORM-2 suspended in 5% human albumin did not significantly affect the anticoagulant properties of the presumed metalloproteinase. The quenching of the inhibitory effect of CORM-2 by albumin indicated that an Ru-based radical formed during carbon monoxide (CO) release from CORM-2, and not the CO released, was responsible for the CORM-2-mediated inhibition of venom anticoagulant activity, as has been published elsewhere in our system [18].

### 2.3. Assessment of the Anticoagulant Activity of Dendroaspis Jamesoni Venom in Human Plasma and of the Inhibitory Effects of EDTA and CORM-2 on Venom Anticoagulant Activity Assessed with Thrombelastography

Similar in the kinetic responsiveness of *D. angusticeps*, the venom of *D. jamesoni* was determined to be best studied via concentration–response trials at a concentration of 1 µg/mL to significantly compromise coagulation but not abolish it. As can be seen in Figure 3, this venom at the indicated concentration did not significantly affect TMRTG values, but it did significantly decrease MRTG and TTG values compared to control conditions. After exposure to EDTA in isolation, the anticoagulant effects of the venom were significantly and completely abolished, suggesting that a metalloproteinase was responsible for the observed anticoagulant properties. Exposure of this venom to 100 µM CORM-2 in isolation in PBS resulted in the loss of its anticoagulant activity, with the TMRTG, MRTG and TTG values of this condition not significantly different from control conditions. Further, the exposure of *D. jamesoni* venom to CORM-2 suspended in 5% human albumin did not significantly affect the anticoagulant properties of the presumed metalloproteinase, with TMRTG values significantly greater than those of the control condition. Lastly, the eradication of the inhibitory effect of CORM-2 by albumin indicated that an Ru-based radical formed during CO release from CORM-2, and not the CO released, was responsible for the CORM-2-mediated inhibition of venom anticoagulant activity.

### 2.4. Assessment of the Anticoagulant Activity of Dendroaspis Viridis Venom in Human Plasma and of the Inhibitory Effects of EDTA and CORM-2 on Venom Anticoagulant Activity Assessed with Thrombelastography

The coagulation kinetic response to *D. viridis* venom was similar to the other two green mamba venoms, but with a few key differences. This venom was found to be optimally studied via concentration–response trials at a concentration of 4 µg/mL to obtain a compromised coagulation but not eliminate clotting activity. As displayed in Figure 4, this venom at the indicated concentration did significantly prolong TMRTG values as well as significantly decrease MRTG and TTG values compared to control conditions. After exposure to EDTA in isolation, the anticoagulant effects of the venom were significantly but only partially diminished, suggestive that a metalloproteinase and another unidentified venom component were responsible for the observed anticoagulant properties. The exposure of this venom to 100 µM CORM-2 in isolation in PBS resulted in the total loss of its anticoagulant activity, with the TMRTG, MRTG and TTG values of this condition not significantly different from control conditions. Subsequently, exposure of the venom to CORM-2 suspended in 5% human albumin did not significantly affect the anticoagulant properties of the unidentified components and metalloproteinase, with TMRTG, MRTG and TTG values significantly greater than those of the control condition or venom exposed to CORM-2 in PBS condition. Last of all, elimination of the inhibitory effect of CORM-2 by albumin indicated that an Ru-based radical formed during CO release from CORM-2, and not the CO released, was responsible for the CORM-2-mediated inhibition of venom anticoagulant activity.

Given the complexity of the data, which are presented as thrombelastographic coagulation kinetic “fingerprints” of the parameters TMRTG, MRTG and TTG, for the convenience of the readership we include Table 2 with a partial summary of our findings.

## 3. Discussion

This investigation achieved its proposed goals. With regard to comparing the general coagulation effects of the four venoms, the venom of the Black Mamba was the least potent of this group by two orders of magnitude. Among the three green mamba venoms, the “all-or-none” phenomena of either not affecting or eliminating coagulation in our system was present, with a very small variation of venom concentration (1–4 µg/mL) yielding conditions conducive of the further testing of coagulation under other conditions. As for the detection of a contribution of metalloproteinases to anticoagulant activity, venom derived from the Black Mamba was not affected by EDTA, suggesting that other venom constituents beyond SVMPs were responsible. In contrast, the venoms of *D. angusticeps* and *D. jamesoni* had their anticoagulant activity extinguished by EDTA exposure, indicating that their anticoagulant activities were mediated by metalloproteinases exclusively. Different in response from the other three venoms, venom obtained from *D. viridis* had anticoagulant activity that was significantly, but only partially, inhibited by EDTA, which supports the concept that another venom constituent was present, that was anticoagulant in nature. Lastly, with regard to the inhibition of anticoagulant activity by CORM-2, black mamba venom displayed no inhibition, whereas all three green mamba venoms had anticoagulant activity significantly and completely inhibited in PBS, but not in the presence of 5% human albumin. As with CORM-2-mediated inhibition of bee venom PLA_2_ anticoagulant activity [18] and the modification of K^+^ channels [20], albumin served as a “histidine sink” that reacts with a proposed Ru-based radical formed from CORM-2 while CO is being released. Our findings with CORM-2 are critical, as the mechanism by which the Ru-based radical effects its modification of proteins is most likely secondary to its binding to histidine residues or thiol groups, as reviewed in our recent work [18]. In sum, with the exception of the venom of the Black mamba, the impact of EDTA- and CORM-2-sensitive anticoagulant activities could be identified in the three green mamba venoms.

The pattern of *Dendroaspis*-venom-mediated anticoagulation is consistent with SVMPs that are fibrinogenolytic, similar to other venoms containing fibrinogenolytic enzymes, wherein TMRTG values may not change, increase a small amount, or by a very great deal compared to plasma without venom addition; MRTG values that always are diminished; and TTG values that are diminished [21,22,23]. The variability in TMRTG is best explained by likely differences in plasma fibrinogen concentration following SVMP enzymatic action, as the time to commencement of clot formation is maintained until fibrinogen concentrations are very small, whereas MRTG and TTG values diminish rapidly with changes in fibrinogen concentrations, as has been documented in human plasma via thrombelastography [24]. Thus, the previously reported results with *Dendroaspis* venoms of “destruction of thromboplastin” and “impaired or delayed thrombin generation” using clotting-based, fibrinogen-dependent methods nearly sixty years ago [11,12] were, to a great extent, detecting SVMP-dependent fibrinogenolysis in the case of the green mambas when considering the data of the present study. However, determining which component of black mamba venom is responsible for the low-potency anticoagulation observed remains to be determined in future investigation.

Thrombelastography has been used to study the hemotoxic effects of snake venoms for several decades, with perhaps the earliest report in English appearing in 1977 [25], followed by just a small number of investigations over the next two decades [26,27,28,29,30] that used whole-blood, platelet-rich plasma and platelet-poor plasma as the model for coagulation kinetic analysis. However, since 2016, there have been numerous reports characterizing/utilizing venoms derived from well over a hundred species with thrombelastography by a few laboratories, with our laboratory producing several of these works. The rationale for utilizing thrombelastography rather than other, standard hemostatic analytical methods is that thrombelastography allows for remarkable plasticity in the amount of thrombin generation, sample type, and information generated compared to other methods. For example, as platelets in whole-blood account for up to 10 times the strength of plasma [31,32], if one wants to determine venom–protein interactions, then platelet poor plasma such as was used in this study is preferable. Further, the ability to differentiate thrombin-like activity (no engagement of factor XIII) from prothrombin-activating/thrombin-generating activity (engagement of factor XIII) between various venoms if facilitated by the known contribution of factor XIII versus fibrinogen alone on clot strength in the plasma via thrombelastography [24,33]. This comparison of the thrombelastographic profiles of thrombin-like and thrombin-generating venoms was recently reviewed by this laboratory [23]. There are several other modifications of either plasma (e.g., heparin addition, calcium omission) or venom in isolation (e.g., EDTA or CORM-2 exposure) that further allow the characterization of whole venom or purified venom enzyme activity, and the readership is encouraged to further review the recent literature on these matters. In sum, thrombelastography is advancing the knowledge of the hemotoxic effects of venoms in a manner that no other modality can.

It is unfortunate that there are no specific inhibitors of Kunitz-type SPI or 3-FTx available to be used to identify what, if any, role these non-enzymatic proteins may have in inhibiting coagulation. Two of the venoms tested, derived from *D. angusticeps* and *D. viridis*, displayed evidence of metalloproteinase-independent anticoagulant properties, with only *D. viridis* demonstrating CORM-2-mediated inhibition of such metalloproteinase-independent anticoagulation. In the absence of an “inhibitor of the inhibitor”, the most direct way to determine which non-enzymatic proteins derived from these are inhibiting coagulation is to separate and purify the proteins—a process beyond the scope of the present investigation. Nevertheless, it may be worth the time and effort to determine whether novel coagulation factor inhibitor properties can be assigned to particular Kunitz-type SPI or 3-FTx for therapeutic anticoagulation purposes.

In conclusion, the present work characterized the anticoagulant properties of the four extant mamba venoms in terms of potency, metalloproteinase activity, and vulnerability to CORM-2 Ru-based radical inhibition. These data continue to add to our mechanistic understanding of how neurotoxic venoms can serve as anticoagulants in vitro and provide a direction for both the inhibition and identification of other non-enzymatic protein anticoagulant agents.

## 4. Materials and Methods

### 4.1. Chemicals and Human Plasma

Lyophilized venoms from *Dendroaspis* species were purchased from Mtoxins, Oshkosh, WI, USA. Venoms were dissolved into calcium-free phosphate-buffered saline (PBS), (Millipore Sigma, Saint Louis, MO, USA) to a concentration of 50 mg/mL, aliquoted, and frozen at −80 °C. Dimethyl sulfoxide (DMSO), tricarbonyldichlororuthenium (II) dimer (CORM-2) and ethylenediaminetetraacetic acid (EDTA) were obtained from Millipore Sigma, Saint Louis, MO, USA. Human albumin solution (5% in 0.9% NaCl) was obtained from Grisfols Biologicals Inc., Los Angeles, CA, USA. Calcium chloride (200 mM) was purchased from Haemonetics Inc., Braintree, MA, USA. Pooled normal human plasma that was sodium citrate anticoagulated and kept frozen at −80 °C was purchased from George King Bio-Medical, Overland Park, KS, USA. This plasma is a commercial product collected from consenting, anonymous and compensated healthy donors by the vendor.

### 4.2. Thrombelastographic Analyses

The volumes of the subsequently described plasma mixtures with various additives all equaled 360 µL. Samples contained 320 µL of plasma; 16.4 µL of PBS, 20 µL of 200 mM CaCl_2_, and 3.6 µL of PBS or venom solution mixture, which were pipetted into a disposable cup in a thrombelastograph^®^ hemostasis system (Model 5000, Haemonetics Inc., Braintree, MA, USA) at 37 °C, and then rapidly mixed by moving the cup up to and then away from the plastic pin five times. The venom was always the last constituent added prior to mixing and data collection. The following viscoelastic parameters, described previously [4,5,6,7,18,19], were measured: time to maximum rate of thrombus generation (TMRTG): this is the time interval (minutes) observed prior to maximum velocity of clot growth; maximum rate of thrombus generation (MRTG): this is the maximum velocity of clot growth (dynes/cm^2^/second); and total thrombus generation (TTG, dynes/cm^2^), the final resistance observed after clot formation is complete. Data were collected for 30 min.

The thrombelastographic paradigm used was performance based, with the criteria that qualifies a venom concentration to be used in experimentation, being that, there are no samples with coagulation parameter values within normal range and that the suppression of coagulation be at least 50% of normal parameter values [4,5,6,7,19]. The initial coagulation concentration-response relationship of each venom was generated over a 1–500 µg/mL range (*n* = 1–2 per concentration) based on previously published concentrations with these venoms [11,12]. After a concentration that satisfied these conditions with each specific venom was determined, experiments were performed as subsequently described.

### 4.3. EDTA Addition Experiments

In experiments with EDTA, the venom was exposed to 5 mM EDTA for 30 min at 37 °C; thereafter, a 1% addition of these EDTA-exposed venom solutions was made to a plasma sample mixture, as outlined in the previous section (Section 4.2) and data were collected until MA was achieved. The final concentration of EDTA in the plasma solution was 50 µM, which would not be expected to affect coagulation following the addition of exogenous calcium to the citrated plasma.

### 4.4. CORM-2 Addition Experiments

In experiments with CORM-2 the conditions utilized were: (1) control condition—no venom, DMSO 1% addition (*v*/*v*) in PBS; (2) Venom condition—venom at the concentration determined in preliminary studies, DMSO 1% addition (*v*/*v*) in PBS; (3) C condition—venom at the concentration as in condition 2, CORM-2 1% addition in DMSO in PBS (100 µM to 1 mM final concentration); (4) C+A condition—venom and CORM-2 1% addition in DMSO in 5% human albumin (100 µM final concentration). Solutions were incubated for 5 min at room temperature, and then 3.6 µL of one of these solutions was added to the plasma sample in the plastic cup.

### 4.5. Graphics and Statistical Analyses

Data are presented as mean ± SD. Graphics were generated with a commercially available program (Origen2020, OrigenLab Corporation, Northampton, MA, USA). Experimental conditions had *n* = six replicates per condition as this provides a statistical power > 0.8 with *p* < 0.05 [4,5,6,7,18,19]. A statistical program utilized one-way analyses of variance (ANOVA) comparisons among the conditions, followed by Holm-Sidak post hoc analysis for between-condition comparisons (SigmaPlot 14, Systat Software, Inc., San Jose, CA, USA). *p* < 0.05 was considered significant.

## Figures and Tables

**Figure 1 ijms-21-02082-f001:**
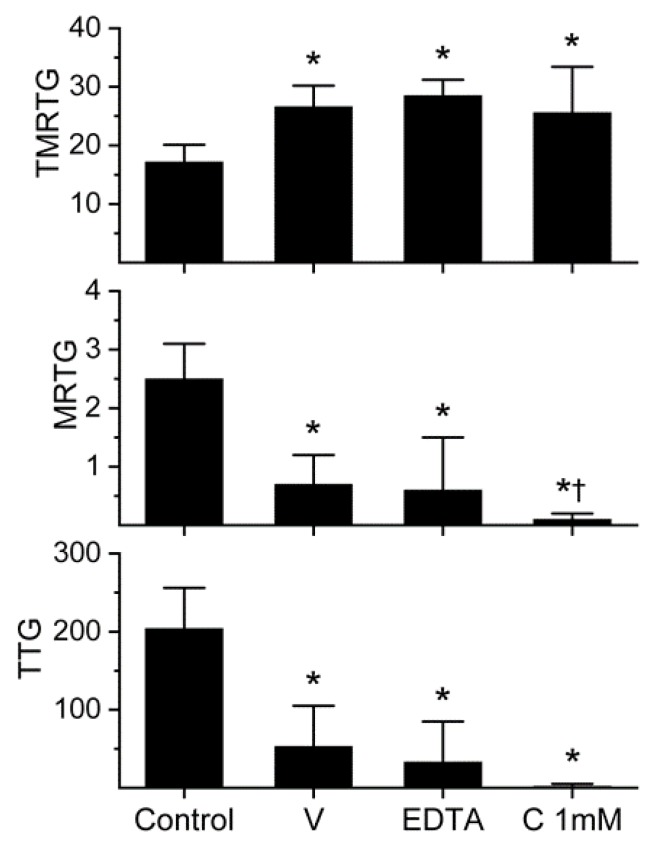
Anticoagulant activity of *D. polylepis* venom in human plasma and effects of EDTA and CORM-2 on venom activity. Data are presented as mean ± SD. TMRTG = minutes; MRTG = dynes/cm^2^/sec; TTG = dynes/cm^2^. Control = no additives; V = 500 µg/mL final venom concentration; EDTA = 500 µg/mL venom after being exposed to 5 mM EDTA; C 1 mM = 500 µg/mL venom after being exposed to 1 mM CORM-2. * *p* < 0.05 vs. control. † *p* < 0.05 vs. V.

**Figure 2 ijms-21-02082-f002:**
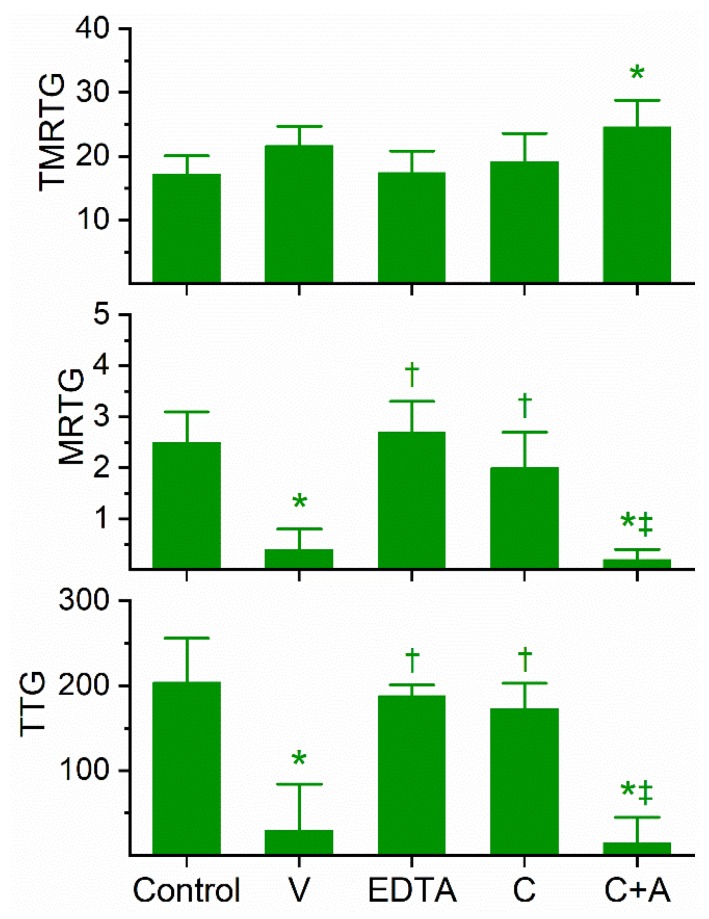
Anticoagulant activity of *D. angusticeps* venom in human plasma and effects of EDTA and CORM-2 on venom activity. Data are presented as mean ± SD. TMRTG = minutes; MRTG = dynes/cm^2^/sec; TTG = dynes/cm^2^. Control = No additives; V = 3 µg/mL final venom concentration; EDTA = 3 µg/mL venom after being exposed to 5 mM EDTA; C = 3 µg/mL venom after being exposed to 100 µM CORM-2 in PBS; C+A = 3 µg/mL venom after being exposed to 100 µM CORM-2 in 5% human albumin. * *p* < 0.05 vs. control; † *p* < 0.05 vs. V; ‡ *p* < 0.05 vs. C.

**Figure 3 ijms-21-02082-f003:**
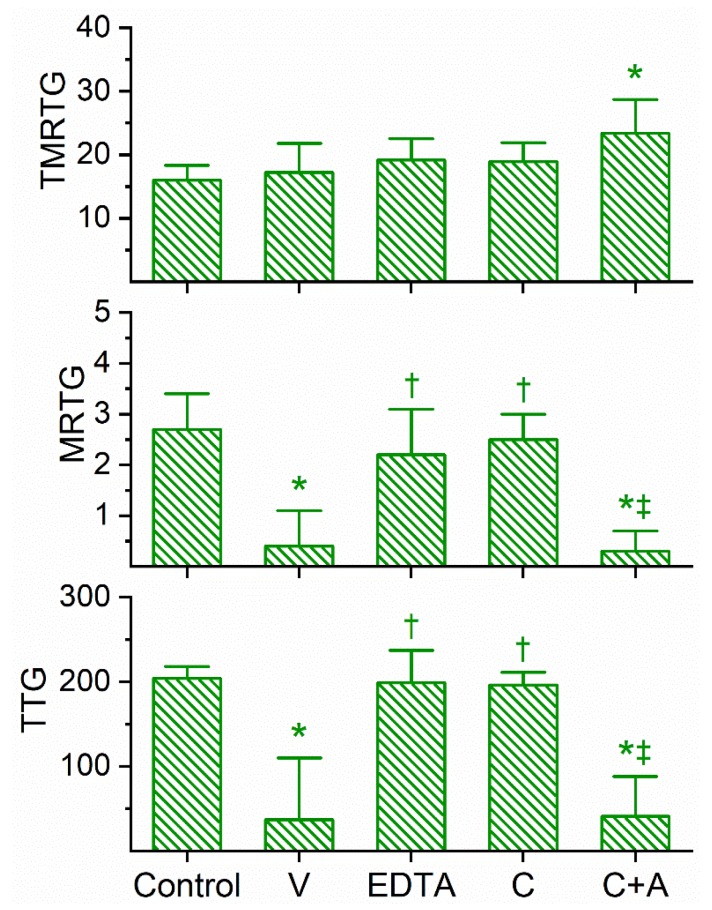
Anticoagulant activity of *D. jamesoni* venom in human plasma and effects of EDTA and CORM-2 on venom activity. Data are presented as mean ± SD. TMRTG = minutes; MRTG = dynes/cm^2^/sec; TTG = dynes/cm^2^. Control = No additives; V = 1 µg/mL final venom concentration; EDTA = 1 µg/mL venom after being exposed to 5 mM EDTA; C = 1 µg/mL venom after being exposed to 100 µM CORM-2 in PBS; C+A = 1 µg/mL venom after being exposed to 100 µM CORM-2 in 5% human albumin. * *p* < 0.05 vs. control; † *p* < 0.05 vs. V; ‡ *p* < 0.05 vs. C.

**Figure 4 ijms-21-02082-f004:**
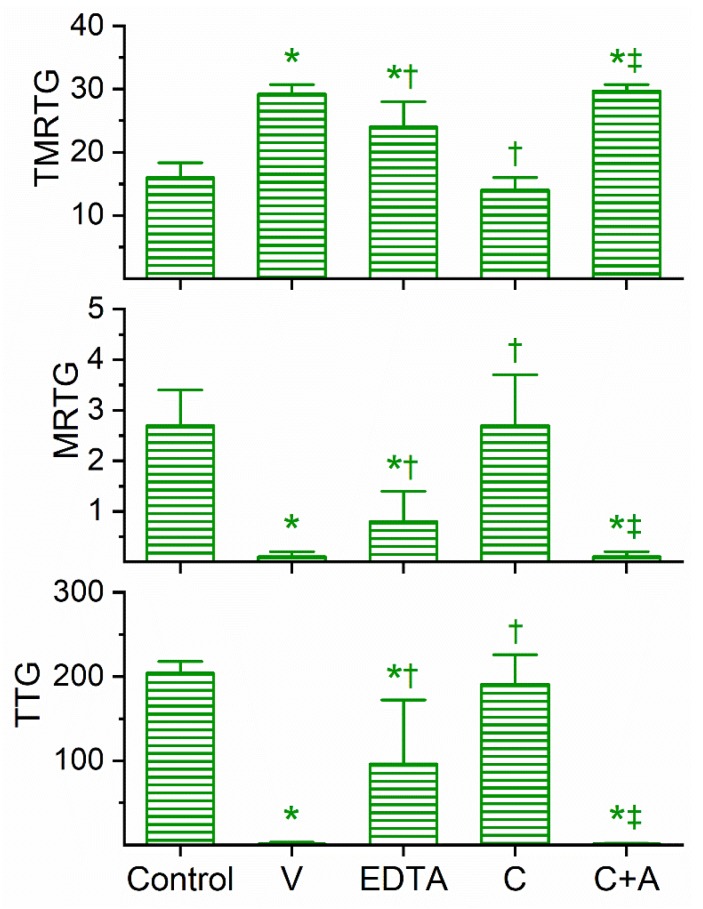
Anticoagulant activity of *D. viridis* venom in human plasma and effects of EDTA and CORM-2 on venom activity. Data are presented as mean ± SD. TMRTG = minutes; MRTG = dynes/cm^2^/sec; TTG = dynes/cm^2^. Control = No additives; V = 4 µg/mL final venom concentration; EDTA = 4 µg/mL venom after being exposed to 5 mM EDTA; C = 4 µg/mL venom after being exposed to 100 µM CORM-2 in PBS; C+A = 4 µg/mL venom after being exposed to 100 µM CORM-2 in 5% human albumin. * *p* < 0.05 vs. control; † *p* < 0.05 vs. V; ‡ *p* < 0.05 vs. C.

**Table 1 ijms-21-02082-t001:** Species and venom proteomes of snake venoms investigated.

Species	3-FTx (%)	Kunitz-Type SPI (%)	Metalloproteinase (%)
*Dendroaspis polylepis* [13,14]	31.0–45.1	48.9–61.1	1.8–3.2
*Dendroaspis angusticeps* [13,15]	69.2–71.4	14.5–16.3	3.4–6.7
*Dendroaspis jamesoni* [13]	80.3	15.1	0.5
*Dendroaspis viridis* [13]	77.7	15.2	2.7

**Table 2 ijms-21-02082-t002:** Summary of the thrombelastographic profile and mechanism of anticoagulant activity.

Species	µg/mL	TMRTG	MRTG	TTG	EDTA Inhibition	CORM-2 Inhibition
*D. polylepis*	500	↑	↓	↓	None	None
*D. angusticeps*	3	No Change	↓	↓	+	+
*D. jamesoni*	1	No Change	↓	↓	+	+
*D. viridis*	4	↑	↓	↓	Partial	+

↑ = venom increased; ↓ = venom decreased; + = inhibition of anticoagulant activity to the point of no significant difference from samples without venom addition; partial = significant inhibition of anticoagulant activity, but still with anticoagulant activity that was significantly different from samples without venom addition.

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
