# Peer review of "Mechanisms Responsible for the Anticoagulant Properties of Neurotoxic Dendroaspis Venoms: A Viscoelastic Analysis"

_ijms, 2020, doi:10.3390/ijms21062082_

Round 1

Reviewer 1 Report

The authors compare the four mamba species to evaluate the anticoagulant properties of their venom using thrombelastography. I have included a pdf with comments in the manuscript directly but have points that should be addressed below;

  1. Thrombelastography is not a technique I have seen in the venom literature and I think the introduction would benefit greatly from a paragraph about its usage history and its utility.
  2. The results would benefit greatly from a restructuring to compare the four species directly in each of the measures rather than how it is presented currently. Instead of having 4 sections (one per species) switch it to three sections (one for TMRTG, MRTG, and TTG) and compare the species to each other for each measure. 
  3. Please have a few people proof-read/edit the writing and sentence structure. I have made some changes within the document but the writing clarity needs to be improved for the manuscript to be ready for publication.

I thank the authors for their contribution and enjoyed reading the manuscript and their investigation of mamba venom.

Author Response

“Suggestions for Authors”

“The authors compare the four mamba species to evaluate the anticoagulant properties of their venom using thrombelastography. I have included a pdf with comments in the manuscript directly but have points that should be addressed below;” 

We thank the reviewer for taking the time to go through and provide us with specific areas and suggestions with the attached pdf.  We have gladly taken most of the suggestions as indicated by highlighted text.  I (VGN) was unaware of the SSAR guidelines concerning common names.  I appreciate your efforts that did educate me personally.

“Thrombelastography is not a technique I have seen in the venom literature and I think the introduction would benefit greatly from a paragraph about its usage history and its utility.” 

We appreciate this comment and now include a brief paragraph as requested in terms of utility of the method and history of thrombelastography in the clinical and laboratory-based analysis of venoms.  However, we ask the reviewer’s indulgence to allow us to place the new paragraph in the Discussion so as to not distract from the flow of prose in the Introduction.  We mention in the paragraph that it appears the first instance of using thrombelastography to analyze venom effects on coagulation appeared in 1977 and include a brief history and utility of the method.  In the past 3-4 years our laboratory and Dr. Bryan Fry’s laboratory has published nearly 50 works that can be found via PubMed or Google, with other investigators joining in as well.  We suspect that thrombelastography will be part of toxinology for years to come.

“The results would benefit greatly from a restructuring to compare the four species directly in each of the measures rather than how it is presented currently. Instead of having 4 sections (one per species) switch it to three sections (one for TMRTG, MRTG, and TTG) and compare the species to each other for each measure.”

            We appreciate this suggestion, but it cannot be validly performed from a statistical standpoint.  The performance-based model that assesses changes in coagulation assure that within venom there is a baseline of coagulopathy that can be compared with the same venom concentration under various biochemical conditions.  What the model does not assure is the exact same degree of hemostatic aberration between venoms.  This is how we and Bryan Fry have looked at over 100 species thus far, and why we present differences by intervention within venom quantitatively and between venom qualitatively.  The presentation of TMRTG, MRTG and TTG by venom is the coagulation kinetic “fingerprint” that we have published many times before, and it is the comparison of such fingerprints that allow us to classify venom activity as anticoagulant, procoagulant, thrombin-like, thrombin-generating, etc.

            While the aforementioned can be a problem, we propose the following to the reviewer.  After all the figures are presented, we have inserted a new table 2 that summarizes the effects of the venoms, concentrations of venom used, and inhibition by EDTA and CORM-2.  We hope that this table will serve to add clarity in lieu of the suggestion made by the reviewer.

“Please have a few people proof-read/edit the writing and sentence structure. I have made some changes within the document but the writing clarity needs to be improved for the manuscript to be ready for publication.”

We have meticulously proof-read the enclosed revision and hope that it is acceptable to the reviewer.

“I thank the authors for their contribution and enjoyed reading the manuscript and their investigation of mamba venom.”

We greatly appreciate this comment from the reviewer and again thank him/her for the thorough review.

Reviewer 2 Report

The manuscript “Mechanisms responsible for the anticoagulant properties of neurotoxic Dendroaspis venoms: a viscoelastic analysis” is aimed to characterize the anticoagulant effects of the venom obtained from four Dendroaspis species by thrombelastography as well as to study the effects of EDTA and CORM-2 on anticoagulant activity of the venoms. Indeed, the authors have investigated anticoagulant effects of four venoms using thrombelastography and found differences in the anticoagulant activity between venoms studied. They also found that anticoagulant activity in three of four venoms was reduced to different extent by the treatment with EDTA or CORM-2. The inhibitory effect of EDTA is usually regarded as evidence for the participation of metalloproteinases in a particular process. This may be the case in this study however phospholipases A2 which may produce anticoagulant effect are inhibited by EDTA as well. The presence of phospholipases A2 in mamba venoms was shown by proteomic studies. To precisely determine the role of metalloproteinases in the anticoagulant mamba effects, the authors should use more specific inhibitor, e.g., batimastat.

On the whole blood thrombelastography provide more comprehensive picture of hemostatic process, however using the plasma it is not clear what are the advantages of the thrombelastography over such standard coagulation tests as thrombin time test, prothrombin time test and partial thromboplastin time test, which allow to determine the coagulation stage affected by the venom. It seems that even not all the data obtained by thrombelastography were analyzed. Thus, D. angusticeps and D. jamesoni venoms did not change TMRTG value, but decreased MRTG and TTG. Does it not mean that these venoms have little effect on the first stages of coagulation pathway and greatly decrease the clot growth rate?

More detailed information about molecular mechanisms of anticoagulant activity of mamba venoms was obtained earlier and described in papers cites by authors (references 11 and 12). Here is a quote from the manuscript “these investigators determined that thrombin generation was impaired, that fibrinogen was digested, that fibrinolysis was impaired, and that platelet aggregation decreased in blood”. Nothing similar has been done in this work. To obtain really new results, the authors should show which stages of coagulation pathways are affected by metalloproteinases whether it will be done by the thrombelastography or the standard coagulation tests.

The application of CORM-2 in this study is not clear. The molecular mechanism of CORM-2 action on the venom components is obscure and its use add nothing to the understanding of the mechanisms responsible for the anticoagulant properties of Dendroaspis venoms.  

Minor points

Lines 44-45. This phrase is very vague. Please, consider revising.

Line 92. “human coagulation”. Please, correct.

Lines 101-102. “the concentration of 500 μg/ml of D. polylepis venom decreased time to maximum thrombus generation” Figure 1 shows that TMRTG was increased after the venom treatment. Please, explain this difference.

Line 109, 148, 171, 197.  “(dashed line)”. There is no dashed line. The same concerns “(solid line)” and “(white inset)” on all four figures.

Lines 130-133. Can the authors explain why “at 4 μg/ml, there was no detectable coagulation”?

Lines 155-157. “Similar in the kinetic responsiveness of D. angusticeps, the venom of D. jamesoni was found to optimally studied via concentration-response trials at a concentration of 1 μg/ml to compromise coagulation but not abolish it.” However, D. angusticeps venom was studied at the concentration of 3 μg/ml, the same concentration is shown in Figure 3.

Lines 179-181. “This venom was found to optimally studied via concentration-response trials at a concentration of 4 μg/ml to compromise coagulation but not eliminate clotting activity in all samples tested.” Figure 4 - 3 μg/ml. It seems, that the capture to figure 2 was copied and pasted without correction to figures 3 and 4.

Lines 216-234. This paragraph is not related to the paper content and should be deleted.

Author Response

“The manuscript “Mechanisms responsible for the anticoagulant properties of neurotoxic Dendroaspis venoms: a viscoelastic analysis” is aimed to characterize the anticoagulant effects of the venom obtained from four Dendroaspis species by thrombelastography as well as to study the effects of EDTA and CORM-2 on anticoagulant activity of the venoms. Indeed, the authors have investigated anticoagulant effects of four venoms using thrombelastography and found differences in the anticoagulant activity between venoms studied. They also found that anticoagulant activity in three of four venoms was reduced to different extent by the treatment with EDTA or CORM-2.”

“The inhibitory effect of EDTA is usually regarded as evidence for the participation of metalloproteinases in a particular process. This may be the case in this study however phospholipases A2 which may produce anticoagulant effect are inhibited by EDTA as well. The presence of phospholipases A2 in mamba venoms was shown by proteomic studies. To precisely determine the role of metalloproteinases in the anticoagulant mamba effects, the authors should use more specific inhibitor, e.g., batimastat.” 

We appreciate this comment by the reviewer, but EDTA does not directly inhibit PLA2.  Snake venom PLA2 tend to be but are not always calcium-dependent, and when the reaction mixture is rendered calcium-free by the addition of EDTA the enzyme loses its ability to digest phospholipids, change platelet aggregation, etc.  However, in our experiments, after the venom was exposed in isolation in calcium-free PBS, it is placed as a 1% (v/v) addition into plasma with calcium added to restore calcium concentrations to normal so as to allow coagulation to commence.  As best as we can find, there is no metal center in PLA2 vulnerable to chelation from EDTA, and as long as there is abundant calcium available, enzymatic activity should commence without problem.  Lastly, based on the indicated reference, we have amended our manuscript to note the very small amount of PLA2 present (zero to less than 0.15%).

This is not the case with SVMP that do typically have a zinc metal center in the catalytic site that is chelated by EDTA and not restored in plasma following calcium addition.  When all this is considered, this is why to the present day that the addition of EDTA to a venom in isolation as an experimental condition is considered specific for identifying SVMP activity inhibition.

“On the whole blood thrombelastography provide more comprehensive picture of hemostatic process, however using the plasma it is not clear what are the advantages of the thrombelastography over such standard coagulation tests as thrombin time test, prothrombin time test and partial thromboplastin time test, which allow to determine the coagulation stage affected by the venom.”

We now include a new paragraph in the Discussion (third paragraph) that points out the renewed utilization of thrombelastography in the toxinology field.  The various modifications and power of the method is also briefly reviewed.  The tests mentioned by the reviewer are insensitive to the effects of fibrinolytic enzymes depending on the degree of thrombin generation – in other words, if thrombin generation in PT and aPTT kinetically outcompete the venom enzymatic degradation of fibrinogen, it will appear as if the venom has no effect except at very large concentration.  Also, in addition to initiation, thrombelastography allows the use of changes in velocity of clot growth and final clot strength to profile enzyme effects on coagulation kinetics.  These and other features unique to thrombelastography are mentioned in the new paragraph.

“It seems that even not all the data obtained by thrombelastography were analyzed. Thus, D. angusticeps and D. jamesoni venoms did not change TMRTG value, but decreased MRTG and TTG. Does it not mean that these venoms have little effect on the first stages of coagulation pathway and greatly decrease the clot growth rate?” 

The reviewer points out a phenomenon often associated with fibrinogenolytic SVMP.  The pattern has more to do with less substrate being available for thrombin to polymerize and likely has nothing to do with inhibition of thrombin generation.  To provide clarity into this matter, we now include a new paragraph in Discussion (second paragraph) to compare and contrast the present study’s findings with earlier works with an emphasis on interpretation of the thrombelastographic data.

“More detailed information about molecular mechanisms of anticoagulant activity of mamba venoms was obtained earlier and described in papers cites by authors (references 11 and 12). Here is a quote from the manuscript “these investigators determined that thrombin generation was impaired, that fibrinogen was digested, that fibrinolysis was impaired, and that platelet aggregation decreased in blood”.  Nothing similar has been done in this work. To obtain really new results, the authors should show which stages of coagulation pathways are affected by metalloproteinases whether it will be done by the thrombelastography or the standard coagulation tests.”

            First, the methodology of references 11 and 12 were at best primitive nearly 60 years ago, with the use of a thrombinoscope (to measure thrombin generation via fluorescence following cleavage of an artificial substrate)  or other devices to really determine if there was a lack of thrombin generation, inhibition of thrombin or an intermediate serine protease, rapid digestion of plasma fibrinogen, or all three.  The mechanisms we identify are SVMP and SVMP-independent anticoagulant activities that may or may not be inhibitable by CORM-2.  We have obtained new and novel results.  We have re-written the passage as follows (lines 49-58):

            ’Using the clotting-based, antiquated technology that was available in the 1960’s, these investigators proposed that thrombin generation was impaired, that fibrinogen was digested, that fibrinolysis was impaired, and that platelet aggregation decreased in blood exposed to black mamba (Dendroaspis polylepis), eastern green mamba (Dendroaspis angusticeps) or Jameson’s green mamba (Dendroaspis jamesoni) venom [11,12].  The fourth extant species, Hallowell’s green mamba (Dendroaspis viridis), was not investigated [11,12].  Of interest, the venom of D. polylepis appeared to be between one and two orders of magnitude less potent as an anticoagulant compared to the other two species tested [11,12].  Critically, the mechanisms responsible for the observed anticoagulant activity in terms of venom compound or enzymes were not addressed by these studies or any subsequent works [11,12].’

            The changes made are highlighted in yellow in the text of our revision.

            Again, we now also include a new paragraph in the Discussion (paragraph 2) to address these issues.

In sum, the authors of [11,12] used methods not publishable today without any identification of venom components responsible for the anticoagulant activity observed.  We did use methods that are modern, frequently published, and we identified or excluded known components of the venoms, differentiating them by species no less. 

“The application of CORM-2 in this study is not clear. The molecular mechanism of CORM-2 action on the venom components is obscure and its use add nothing to the understanding of the mechanisms responsible for the anticoagulant properties of Dendroaspis venoms.”

            Again, we must respectfully disagree with the reviewer.  We clearly outlined the rationale for the use of CORM-2 in our goals in the Introduction: “ Such determinations of the mechanism responsible for CORM-2 mediated inhibition of snake venom hemotoxicity is of interest considering its in vitro efficacy against numerous species’ venoms or isolated venom enzymes as previously noted [4-7,19]. Given that the Ru-based reactive species formed by CORM-2 reacts with both histidine and thiol groups as reviewed in our previous work, there are putative molecular targets on the compounds contained within the venoms tested.  Thus, it was our express purpose to assess the effects of CORM-2 on anticoagulant activity as outlined in our goals, and there is a molecular basis for these assessments.   

Minor points

“Lines 44-45. This phrase is very vague. Please, consider revising.”

We appreciate this comment and have modified the phrase cited.

“Line 92. “human coagulation”. Please, correct.” 

The phrase conveyed our meaning, but we are happy to add the words ‘whole blood and plasmatic’ to the text.

“Lines 101-102. “the concentration of 500 μg/ml of D. polylepis venom decreased time to maximum thrombus generation” Figure 1 shows that TMRTG was increased after the venom treatment. Please, explain this difference.” 

We thank the reviewer for detecting this error.  Compromised coagulation always has increased TMRTG, decreased MRTG and decreased TTG.  The text has been corrected to reflect this.

“Line 109, 148, 171, 197.  “(dashed line)”. There is no dashed line. The same concerns “(solid line)” and “(white inset)” on all four figures.” 

We appreciate this comment and have corrected all four figure captions.

“Lines 130-133. Can the authors explain why “at 4 μg/ml, there was no detectable coagulation”?” 

The thrombelastographic model which we have used for a few years to assess the effects of venoms on coagulation is performance based.  The criteria concerning the compromise of coagulation needed is indicated in the text.  The lines cited by the reviewer are: “Specifically, at a concentration of 2 µg/ml, there was no detectable compromise of coagulation; in contrast, at 4 µg/ml, there was no detectable coagulation.”  The simplest explanation for no detectable coagulation was that the enzyme/compound with anticoagulant activity in the venom was able to destroy its target molecule faster than thrombin generated from the contact protein mediated generation of thrombin could polymerized fibrinogen and activate factor XIII to polymerize fibrin polymers that could engage the cup and pin of the thrombelastograph and be translated into viscoelastic data.

            “Lines 155-157. “Similar in the kinetic responsiveness of D. angusticeps, the venom of D. jamesoni was found to optimally studied via concentration-response trials at a concentration of 1 μg/ml to compromise coagulation but not abolish it.” However, D. angusticeps venom was studied at the concentration of 3 μg/ml, the same concentration is shown in Figure 3.”

            All text and figure legends have been reassessed and modified as needed to make sure that all venom concentrations are correctly depicted for each venom.

“Lines 179-181. “This venom was found to optimally studied via concentration-response trials at a concentration of 4 μg/ml to compromise coagulation but not eliminate clotting activity in all samples tested.” Figure 4 - 3 μg/ml. It seems, that the capture to figure 2 was copied and pasted without correction to figures 3 and 4.” 

There is no correction to make, as the procedure to determine what concentration of each venom would meet the a priori conditions was followed.  We have modified the indicated sections with different text that conveys the same meaning.

“Lines 216-234. This paragraph is not related to the paper content and should be deleted.” 

We respectfully disagree with the reviewer’s comment.  The importance of the use of CORM-2 as a source of potential inhibition via its Ru-based radical as a diagnostic tool and potential therapeutic agent is grounded in the literature we have generated over the past three years.  In the spirit of compromise, we have removed some of the lines concerning the stoichiometry of human albumin as a histidine-rich molecule to implicate the Ru-based radical as the mechanism by which CORM-2 inhibits anticoagulant activity.  These lines have been replaced with further statements concerning the utilization of CORM-2 for mechanistic purposes.

Round 2

Reviewer 1 Report

Thank you for the revisions. The additional paragraphs to the discussion and the addition of Table 2 improve the clarity of the manuscript.

Line 206 - second arrow should say "decreased".

Well done.

Reviewer 2 Report

All my comments are addressed in great details. There are no other questions.